# ^1^H-NMR-Based Metabolomics in Autism Spectrum Disorder and Pediatric Acute-Onset Neuropsychiatric Syndrome

**DOI:** 10.3390/jcm11216493

**Published:** 2022-11-01

**Authors:** Antonella Gagliano, Federica Murgia, Agata Maria Capodiferro, Marcello Giuseppe Tanca, Aran Hendren, Stella Giulia Falqui, Michela Aresti, Martina Comini, Sara Carucci, Eleonora Cocco, Lorena Lorefice, Michele Roccella, Luigi Vetri, Stefano Sotgiu, Alessandro Zuddas, Luigi Atzori

**Affiliations:** 1Child & Adolescent Neuropsychiatry Unit, Department of Biomedical Sciences, “A. Cao” Paediatric Hospital, University of Cagliari, 09121 Cagliari, Italy; 2Department of Health Science, “Magna Graecia” University of Catanzaro, 88100 Catanzaro, Italy; 3Clinical Metabolomics Unit, Department of Biomedical Sciences, University of Cagliari, 09042 Cagliari, Italy; 4Faculty of Health and Medical Sciences, University of Surrey, Guildford GU2 7XH, UK; 5Multiple Sclerosis Regional Center, ASSL Cagliari, Department of Medical Sciences and Public Health, University of Cagliari, 09126 Cagliari, Italy; 6Multiple Sclerosis Regional Center, ASSL Cagliari, 09126 Cagliari, Italy; 7Department of Psychology, Educational Science and Human Movement, University of Palermo, 90128 Palermo, Italy; 8Oasi Research Institute-IRCCS, Via Conte Ruggero 73, 94018 Troina, Italy; 9Child Neuropsychiatry Unit, Department of Medicine, Surgery and Farmacy, University of Sassari, 07100 Sassari, Italy

**Keywords:** autism spectrum disorder, pediatric acute-onset neuropsychiatric syndrome, metabolomics, biomarkers’ evaluation, pathways analysis

## Abstract

We recently described a unique plasma metabolite profile in subjects with pediatric acute-onset neuropsychiatric syndrome (PANS), suggesting pathogenic models involving specific patterns of neurotransmission, neuroinflammation, and oxidative stress. Here, we extend the analysis to a group of patients with autism spectrum disorder (ASD), as a consensus has recently emerged around its immune-mediated pathophysiology with a widespread involvement of brain networks. This observational case-control study enrolled patients referred for PANS and ASD from June 2019 to May 2020, as well as neurotypical age and gender-matched control subjects. Thirty-four PANS outpatients, fifteen ASD outpatients, and twenty-five neurotypical subjects underwent physical and neuropsychiatric evaluations, alongside serum metabolomic analysis with 1H-NMR. In supervised models, the metabolomic profile of ASD was significantly different from controls (*p* = 0.0001), with skewed concentrations of asparagine, aspartate, betaine, glycine, lactate, glucose, and pyruvate. Metabolomic separation was also observed between PANS and ASD subjects (*p* = 0.02), with differences in the concentrations of arginine, aspartate, betaine, choline, creatine phosphate, glycine, pyruvate, and tryptophan. We confirmed a unique serum metabolomic profile of PANS compared with both ASD and neurotypical subjects, distinguishing PANS as a pathophysiological entity per se. Tryptophan and glycine appear as neuroinflammatory fingerprints of PANS and ASD, respectively. In particular, a reduction in glycine would primarily affect NMDA-R excitatory tone, overall impairing downstream glutamatergic, dopaminergic, and GABAergic transmissions. Nonetheless, we found metabolomic similarities between PANS and ASD that suggest a putative role of N-methyl-D-aspartate receptor (NMDA-R) dysfunction in both disorders. Metabolomics-based approaches could contribute to the identification of novel ASD and PANS biomarkers.

## 1. Introduction

Pediatric acute-onset neuropsychiatric syndrome (PANS) is a heterogeneous disorder characterized by the acute onset of obsessive-compulsive disorder and/or severe eating restriction, with at least two concomitant cognitive, behavioral, affective, or somatic symptoms (e.g., anxiety, irritability/depression [1,2], sensorimotor abnormalities, and enuresis). PANS onset is thought to be triggered by post-infectious immune mechanisms [3,4,5,6,7] involving the release of neuroinflammatory mediators within the central nervous system (CNS) [2,8]. Currently, PANS remains a controversial entity because of its heterogeneous presentation, complex constellation of psychiatric and/or somatic and motor symptoms, and its intrinsic autoimmune/inflammatory nature [9].

Autism spectrum disorder (ASD) is a neurodevelopmental disorder with shared genetic and environmental influences. Although several pieces of the aetiological puzzle of ASD are still missing, a consensus has recently emerged around its immune-mediated pathophysiology with a widespread involvement of brain networks [10,11].

On the other hand, genetic risk factors, leading to dysregulation of immune pathways, could play a role in PANS. A very recent paper [12] identified ultra-rare variants in 11 genes, separable into two broad functional categories regulating both the peripheral immune responses and the brain microglia (PPM1D, CHK2, NLRC4, RAG1, and PLCG2) and the synaptogenesis (SHANK3, SYNGAP1, GRIN2A, GABRG2, CACNA1B, and SGCE). These last genes are involved in the pathogenesis of neurodevelopmental disorders and, in particular, of ASD [13], suggesting a multifactorial and probabilistic role of genetic and environmental factors (different kinds of stressors such as infections) in both PANS and ASD.

Several commonalities between ASD and PANS stem from overlapping clinical and immunological features, which result in a more challenging diagnostic path. Therefore, specific biomarkers are desirable to support clinicians in the diagnostic process.

In this scenario, metabolomics could offer the opportunity to screen new biomolecules as potential biomarkers as it allows the quantification of an array of metabolites through two sensitive and specific methodologies: nuclear magnetic resonance (NMR) [14] and mass spectrometry (MS) [15]. Attempts to identify metabolomic biomarkers in several distinct brain disorders have revealed intriguing results [14,15,16,17,18].

As for ASD, serum and urine metabolomics have identified biomarkers attributable to gut dysmicrobism, amino acid metabolism, and mitochondrial dysfunction [19,20] while, regarding PANS and metabolomics, this field appears to not be sufficiently explored, with little reported evidence [21,22].

With increasing consideration given to immune-mediated hypotheses of neurodevelopmental disorders such as ASD [22], we aimed to compare the serum metabolic profile of patients affected by PANS and ASD to find specific fingerprints that could be useful in the classification of the patients and to explore pathophysiological mechanisms still unclear for both disorders. 

## 2. Materials and Methods

### 2.1. Study Design

We collected serum samples of patients affected by PANS and ASD and healthy subjects. Firstly, we compared the hydrophilic metabolic profile (including amino acids, sugars, biogenic amines, fatty acids, and organic acids) of the ASD patients to the control class to evidence a specific pathological pattern with respect to a normal condition. Then, we compared the ASD and PANS metabolic profiles to find common and specific features that could be useful in the correct classification of the patients and to explore the specific pathophysiological aspects of the two conditions. We also planned to evaluate a possible linear correlation between the metabolic profile and psychodiagnostic scales.

### 2.2. Participants and Ethical Aspects

The study was conducted in accordance with the Declaration of Helsinki [23] and was approved by the Cagliari University Hospital Ethics Committee. Informed consent for study participation and data publication was obtained from patients’ parents or legal guardians. Patients and controls were enrolled from June 2019 to May 2020 at the outpatient service of the Child and Adolescent Neuropsychiatric Unit, “G. Brotzu” Hospital Trust, Cagliari.

Diagnosis of PANS was confirmed by two child psychiatrists based on NIMH 2010 criteria [2]. Patients with ASD were diagnosed according to DSM-5 criteria and subject to the Autism Diagnostic Observation Schedule-Second Edition (ADOS-2). The control group (HC) included 25 neurotypical children living in the same geographic area and matched for age and gender.

Extensive physical, neurological, and psychiatric examinations were performed, as well as laboratory tests including a complete blood count, renal and liver function testing, mineral panel, thyroid indices, and inflammation markers to exclude metabolic or systemic diseases.

The exclusion criteria were as follows: (I) autoimmune diseases or cancer; (II) other medical or neurological/psychiatric diseases; (III) active treatment with steroids or non-steroidal anti-inflammatory drugs; and (IV) a lack of written informed consent from parents or legal guardian or withdrawn consent from patients themselves.

### 2.3. Psychiatric Evaluation

All patients enrolled were studied by a large panel of standardized scales and questionnaires to assess symptoms and clinical severity: the Pediatric Anxiety Rating Scale (PARS), Pediatric Acute Neuropsychiatric Symptom Scale (PANSS), Children’s Yale-Brown Obsessive Compulsive Scale (CYBOCS), Yale Global Tic Severity Scale Score (YGTSS), Children’s Global Assessment Scale (C-GAS), Universidade Federal de Minas Gerais Sydenham’s Chorea Rating Scale (USCRS), and Full Scale Intelligence Quotient (FSIQ) assessed by Wechsler Intelligence Scale for Children (4th Edition) (WISC-IV). The results are reported in the Appendix A.

### 2.4. Sample Preparation and Data Analysis

Blood samples (10 mL by venipuncture) were collected after an overnight fast (12 h) and centrifuged at 2500× *g* for 10 min at 4 °C. Sera, stored at −80 °C until analysis, were analysed as described in our previous study (Murgia et al., 2021) [18]. Details are reported in the Appendix A. 

## 3. Results

Among 119 consecutive outpatients referred for PANS (*n* = 52) and ASD (*n* = 67), 70 were excluded (mostly for active treatment with psychiatric or anti-inflammatory drugs). Therefore, 34 PANS and 15 ASD outpatients were recruited. Demographic details are reported in Table 1. More in detail, the number of subjects for each class, the percentage of females and males, and data about their age (mean, standard deviation, and range) are reported.

The ^1^H-NMR analysis allowed the identification of 44 hydrophilic metabolites, including amino acids, fatty acids, sugars, and biogenic amines. 

Initially, principal component analysis (PCA) was performed using the whole bins dataset (ASD, PANS, and controls sample). Hotelling’s T2 test identified one strong PANS outlier, which was ruled out (Appendix A). Subsequently, supervised models (PLS-DA and OPLS-DA) were performed to compare firstly ASD with HC and then PANS with ASD classes (Figure 1). Separation of the samples consistent with clinical diagnoses was observed and the models were validated with the respective permutation test (Table 2).

As the ASD group was composed entirely of male subjects, a novel comparison was conducted by excluding females from HC and PANS groups to test for gender-driven biases, although separation resulted from identical variables (Appendix A). 

The most relevant variables were identified for each model through the volcano plot analysis and the corresponding VIP-value. Variables with VIP > 1 were identified and subject to univariate analysis with Mann–Whitney U-test. 

With a cut-off of *p* < 0.05, asparagine, aspartate, betaine, glycine, lactate, glucose, and pyruvate showed the greatest differences between ASD and HC, while arginine, aspartate, betaine, choline, creatine phosphate, glycine, pyruvate, and tryptophan exhibited the largest differences between PANS and ASD groups (Figure 2).

Subsequently, these metabolites were selected for receiver operating characteristic (ROC) curve analysis (Table 3). 

MetaboAnalyst was used to characterize both altered pathways and enrichment analyses in each group. When comparing ASD and HC, glycolysis/gluconeogenesis balance, pyruvate, alanine, aspartate and glutamate, glycine, and serine and threonine metabolism were the most altered (Figure 3A,B). When comparing PANS and ASD, glycine, serine and threonine metabolism, arginine biosynthesis, arginine and proline, alanine, aspartate and glutamate metabolism, the urea cycle, and ammonia recycling were the most altered (Figure 4A,B).

Finally, to test a possible linear correlation between the metabolic profile and psychodiagnostic scales (PARS, PANSS, CYBOCS, YGTSS, C-GAS, WISC-IV, and USCRS), PLS multivariate models were performed for PANS and ASD classes (Figure 5). 

YGTSS was omitted for the ASD class because only two patients exhibited concurrent tic disorders. PLS showed weak correlations for all of the Y parameters in the PANS group, except for the PANSS scale, while strong correlations were identified between the metabolomic profile of ASD metabolomic and clinical scores of C-GAS, WISC-IV, and USCRS (Table 4).

## 4. Discussion

The aim of the present study was to investigate similarities and specificities in the metabolomic signature of two developmental neuropsychiatric disorders (i.e., ASD and PANS), showing different clinical features, but potentially similar physiopathological mechanisms.

The results of the present study show that metabolomic fingerprint was detectable in ASD subjects compared with HC. In particular, asparagine, aspartate, betaine, glycine, lactate, glucose, and pyruvate exhibited the greatest differences. Conversely, arginine, aspartate, betaine, choline, creatine phosphate, glycine, pyruvate, and tryptophan are the most discriminant metabolites between ASD and PANS (Figure 2). 

Considering the common and the specific metabolic features of ASD and PANS classes, we can summarize that asparagine and glycine appeared to be significantly decreased in both ASD and PANS sera and that impaired glucose metabolism appears to be a key feature of ASD, while a decreased tryptophan concentration, previously identified as a significant feature of metabolomic profile of PANS [18], was not observed in ASD subjects.

### 4.1. Decrease in Glycine (Gly) and Asparagine Concentrations Is a Shared Biomarker of ASD and PANS

In accordance with other reports [24,25], in our previous work, we found Gly serum levels to be significantly lower in PANS [18] than in healthy controls. Interestingly, now, we found an even greater distinction between ASD patients and healthy subjects. The reduction in Gly bioavailability in ASD subjects is supposed to be related to malabsorption induced by gut dismicrobism and the inflammatory disruption of the intestinal barrier [26,27]. Analogous mechanisms involving inflammation and changes to the intestinal microbiota leading to reductions in Gly concentration have also been suggested in PANS [28]. Through the inhibition of the nuclear factor kappa B (NF-kB) pathway and the synthesis of pro-inflammatory cytokines (IL-6, TNF-α, and IL-8), glycine has both in vitro and in vivo anti-inflammatory properties [29]. Thus, its reduced concentration could contribute to a pro-inflammatory state in both PANS and ASD [30].

It should be considered, however, that an increase in urinary Gly has been found in distinct samples of autistic children [31,32], suggesting that Gly regulation may be more complex than solely gut bacterial alterations and inflammation. During neurodevelopment, Gly receptors promote the spontaneous activity of striatal medium spiny neurons and support the maturation of glutamatergic inputs [33]. Rare variants in the alpha 2 subunit of the Glycine receptor have been observed in some autistic subjects [34], confirming a possible functional role for abnormal glycinergic signaling in autism [35]. 

Gly is a pivotal neurotransmitter in several human psychiatric disorders and experimental models [36,37]. ^1^H-NMR analysis showed that Gly was significantly decreased in the plasma and urine of neuroleptic-naïve schizophrenia patients at their first outset. After 6 weeks of risperidone therapy, serum Gly increased in parallel with symptomatic amelioration, presumably owing to its potentiating effect on N-methyl-D-aspartate receptors (NMDA-R) [38] and modulation of dopaminergic signaling [39]. Similarly, low serum Gly was found in untreated patients with major depressive disorder, with the clinical improvement associated with increasing Gly levels [40]. Alterations to glycinergic transmission were also described in a metabolomic characterization of an anxiety-trait mouse model [41]. Being central to the regulation of locomotor behavior and related disorders such as Tourette disorder [42] and impaired glycinergic tonic inhibition in crucial brain regions [43], Gly is widely distributed in prefrontal and limbic cortices and has been related to temporal lobe epilepsy and ASD [44]. Gly exerts most of its biological effects through co-agonism of glutamate in binding excitatory cation-selective NMDA-R, widely distributed in CNS [39,45]. Consequently, it is feasible that a reduction in available glycine would primarily affect the NMDA-R excitatory tone, overall impairing downstream glutamatergic, dopaminergic, and GABAergic transmissions [39]. NMDA-R dysfunction at the hippocampal level could contribute to cognitive disturbances or the so-called “brain fog” phenomenon recently described in PANS subjects [46]. Here, the Gly metabolomic profile was strongly correlated with the cognitive evaluation in ASD (Table 3), reinforcing the idea that glycine imbalance could be related to cognitive impairment.

In our study, asparagine concentration also appeared to be significantly decreased in both ASD and PANS [18]. As asparagine and aspartate concentrations trended in opposite directions in ASD when compared with controls, it is plausible that a reduction in asparagine concentration is independent of aspartate metabolism. Asparagine is associated with the inhibition of inflammation, cell growth, and autophagy [47,48]. An asparagine deficit in both ASD and PANS could lead to detrimental effects on inflammatory homeostasis, suggesting its potential “inflammatory signature” for both groups. It should be considered, however, that the increase in serum asparagine in ASD patients has also been reported [49,50]; distinct ethnic backgrounds of the patients or different measurement methodologies (i.e., HPLC) could explain these differences.

### 4.2. Impaired Glucose Metabolism Appears to Be the Key Feature of ASD 

To date, metabolomic literature relating to ASD mainly consists of studies examining urine samples, showing perturbations in amino-acid-related and mitochondrial metabolic pathways [19]. We found decreased glucose and pyruvate concentrations and increased lactate as crucial and specific fingerprints of ASD, in keeping with other blood-based studies (i.e., glucose-alanine and urea cycle dysfunction) [51]. Glucose dysfunction (i.e., decreased plasma glucose and increased alanine and lactate) was found in other neuropsychiatric conditions such as neuroleptic-naïve schizophrenia, suggesting that increased glycolysis and/or cellular uptake of these metabolites may be a common feature of a multitude of psychiatric disorders [38]. [18F]-fluorodeoxyglucose PET studies in ASD show many CNS areas with decreased metabolic rates when compared with controls [52]. Very recently, decreased insulin sensitivity was found in an ASD cohort compared with HC [53], suggesting that insulin resistance could be primarily located in highly insulin-sensitive brain areas, thereby reducing neuronal glucose uptake and causing mitochondrial dysfunction, lactate overproduction, increased oxidative stress, and reduced availability of neuroprotective factors such as BDNF [54]. 

### 4.3. Decreased Tryptophan Concentration Is a Relevant Feature of PANS, but Not of ASD

Previously, we reported significantly lower serum tryptophan levels in PANS patients than controls, hypothesizing a pathophysiological role for tryptophan metabolism in PANS [18], as in other psychiatric disorders [55,56,57], including, though controversially, ASD [58,59]. In the present study, we found no differences in tryptophan concentration between ASD and HC, suggesting that decreased serum tryptophan could be indicative of a discriminating pathophysiological facet of PANS. 

Aside from 5-HT and melatonin biosynthesis, the predominant pathway of tryptophan metabolism in humans is the kynurenine pathway, which is upregulated under infectious and/or inflammatory conditions, where as much as 60% of the CNS kynurenine pool originates from the blood [60]. Indeed, kynurenine metabolites can cross the blood–brain barrier and enter the CNS [61,62], with neurotoxic effects, inducing microglial activation and neuroinflammation [63], contributing to the outset of symptoms. Moreover, kynurenine itself exerts pro-apoptotic action during immune activation, decreasing the T-helper 1 (Th1) pool and the cross-inhibitory effect on Th2 cell differentiation [64].

Experimentally, a PANS-like phenotype can be induced in rats by inoculating the striatum with quinolinic acid, a kynurenine metabolite [65]. Quinolinic acid exerts neurotoxic effects through excessive NMDA-R agonism, reactive oxygen species (ROS) formation, lipid peroxidation, mitochondrial damage, and apoptosis in neurons, astrocytes, and oligodendrocytes, which, in turn, trigger an inflammatory state [66,67]. Finally, as 5-HT is a precursor of melatonin, tryptophan deficiency could be an upstream contributory factor to sleep disturbances (e.g., ineffective sleep, periodic limb movements, REM-sleep without atonia, insomnia, and parasomnias), enumerated among PANS symptoms [46,68]. 

As reported and discussed in a previous paper [18], and concurrently with the existing literature, tryptophan metabolism may play a central role in the pathogenesis of affective, motor, and cognitive symptoms more in PANS than in ASD subjects. 

## 5. Conclusions and Limitations of the Study

Biomarkers may represent both diagnostic predictors and key pathophysiological targets, potentially playing a crucial role in defining innovative therapeutic approaches. Through ^1^H-NMR metabolomics, we “sampled” a downstream biological system at the common final pathway level to overcome the complexity of the upstream etiological events (e.g., infections and autoantibodies).

The main limitation of this approach is the significant clinical and biological heterogeneity of subjects with ASD [69,70]. Molecular genetics studies have identified a few hundred ASD risk genes [71,72] that may significantly amplify the difficulties in identifying reliable biomarkers for the disorder. ASD susceptibility genes, however, appear to converge in a discrete number of biological pathways [72]: several lines of evidence suggest that many of these biological pathways (and thus many genes) are shared, at least in children and adolescents, among different psychiatric disorders [73,74,75,76]. Interestingly, many genes implicated in ASD appear to converge into classical cytokine signalling pathways, suggesting the presence of an immunological-inflammatory environment in the pathogenesis of ASD [72,77].

A recent study performed on 516 very young ASD children (18 to 48 months of age) evidenced specific metabotypes detected in different subgroups of ASD, measuring the ratio between glutamine, glycine, and ornithine, and the branched-chain amino acids (BCAAs), which appeared to be a key feature of the ASD disease [78]. We did not identify alteration to BCAA metabolism; we recruited a relatively small sample of patients, which did not allow for stratification depending on the ASD subtype. 

However, our results are in line with former evidence, suggesting PANS and, at least in part, ASD as immune-mediated disorders, although other biological mechanisms may play crucial roles in both disorders.

Further investigations are needed to clarify the translational value of amino acids’ serum level in the pathophysiology of brain dysfunction and their role in ASD and in PANS. Consistently with previous studies on amino acid dysregulation and the stratification of ASD patients based on the identification of “metabotypes” [78], our study lays the foundation for discovering metabolic tests facilitating the PANS diagnosis. For both ASD and PANS, the identification of “metabotypes” could bring the research towards the discovery of new targeted therapeutic interventions.

Furthermore, both analogous and diverse mechanisms of glucose dysfunction in ASD and other overlapping neuropsychiatric disorders such as schizophrenia should be addressed (e.g., whether central or peripheral and whether involving insulin or counter-regulatory hormones). 

Finally, a larger ASD cohort and the use of mass spectrometry to identify undetected metabolites (lipophilic compounds) are warranted to support the present findings and interpretation.

## Figures and Tables

**Figure 1 jcm-11-06493-f001:**
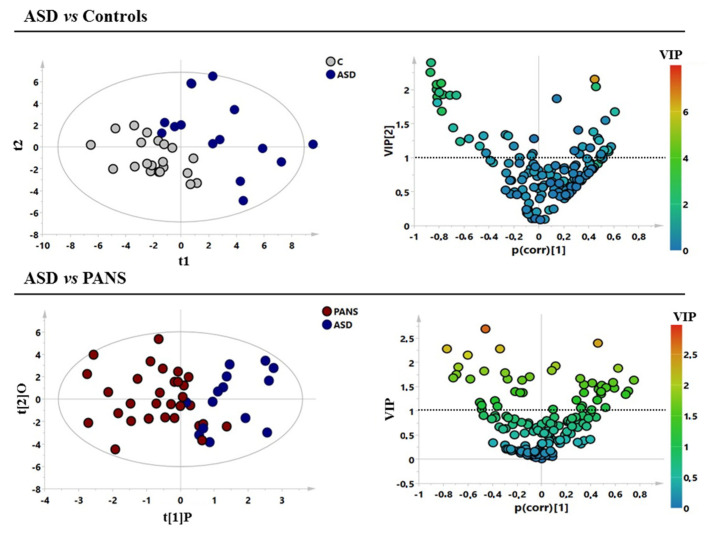
Supervised models of the analysed classes. (**A**,**B**) ASD (blu circles) vs. control subjects (grey circles) with the respective volcano plot. (**C**,**D**) PANS (red circles) vs. ASD (blue circles) patients with the respective volcano plot. ASD = autism spectrum disorder; PANS = pediatric acute-onset neuropsychiatric syndrome.

**Figure 2 jcm-11-06493-f002:**
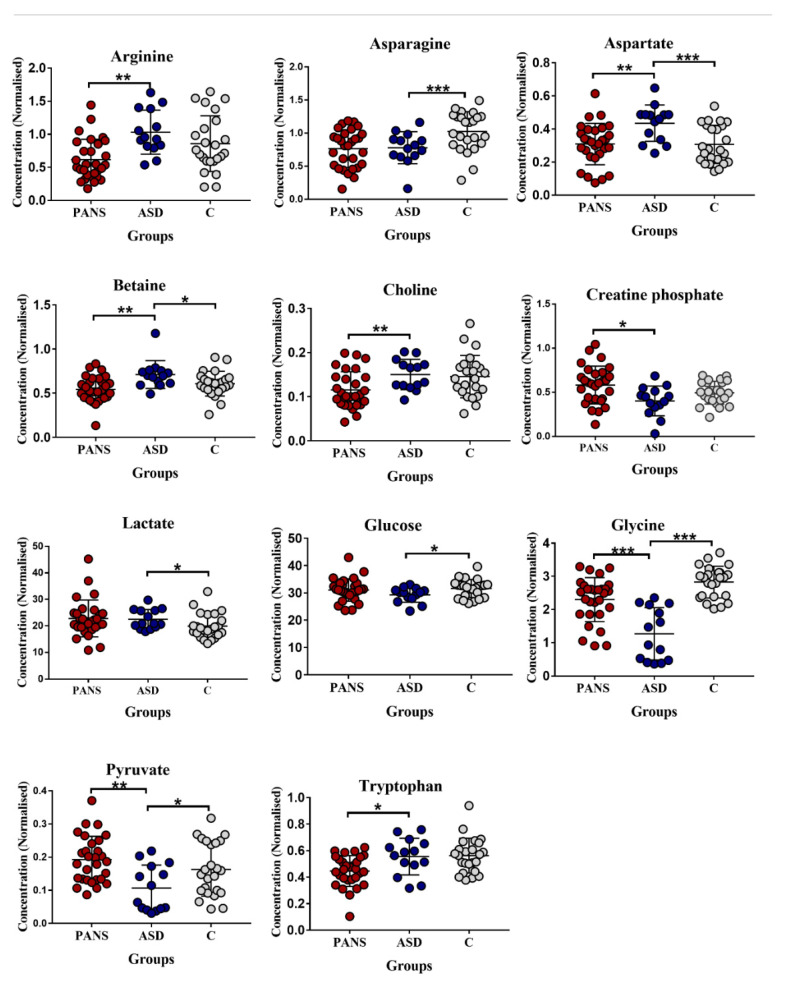
Comparison between controls and ASD and PANS patients. Bar graphs of the metabolites exhibit a *p*-value of <0.05 (Mann–Whitney U-test). Grey dots represent the control class, red dots represent the PANS class, while blue dots represent the ASD patients. ASD = autism spectrum disorder; PANS = pediatric acute-onset neuropsychiatric syndrome. * *p* < 0.05, ** *p*-value < 0.01, *** *p*-value < 0.001.

**Figure 3 jcm-11-06493-f003:**
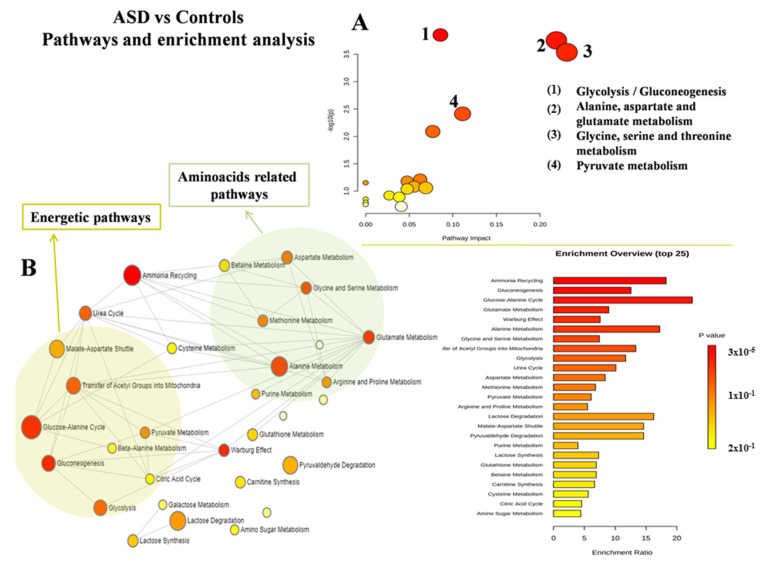
Metabolic pathways’ analysis (**A**) and enrichment analysis (**B**) of the comparison between control subjects and ASD patients. ASD = autism spectrum disorder.

**Figure 4 jcm-11-06493-f004:**
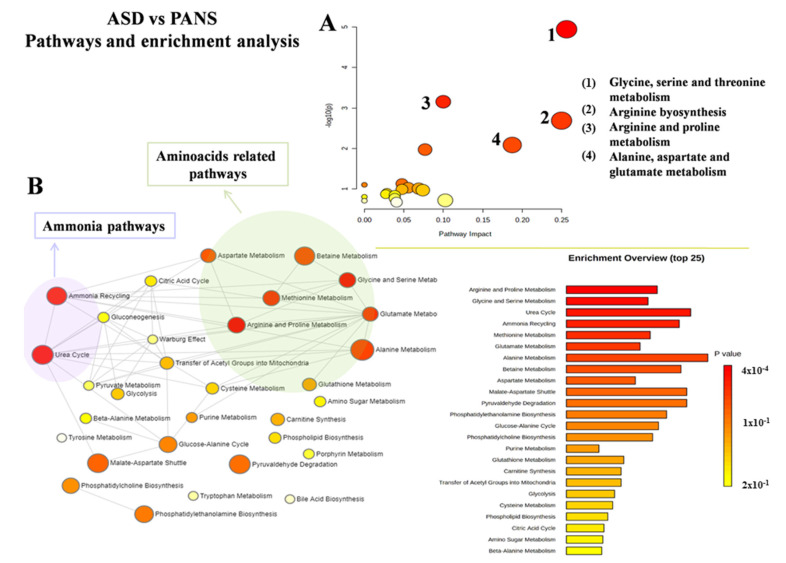
Metabolic pathways’ analysis (**A**) and enrichment analysis (**B**) of the comparison between ASD (autism spectrum disorder) and PANS (pediatric acute-onset neuropsychiatric syndrome) patients.

**Figure 5 jcm-11-06493-f005:**
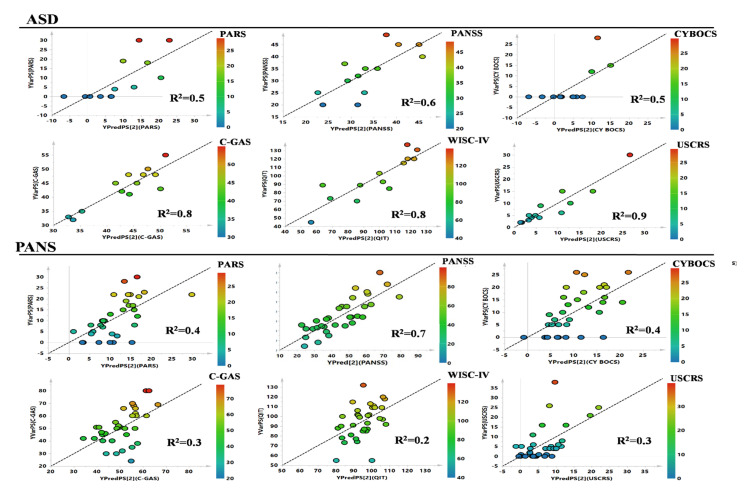
PLS correlation analysis between the metabolic profiles of the three classes of patients and clinical parameters. Clinical parameters were assessed by PARS (Pediatric Anxiety Rating Scale), PANSS (Pediatric Acute Neuropsychiatric Symptom Scale), CYBOCS (Children’s Yale-Brown Obsessive Compulsive Scale), YGTSS (Total Yale Global Tic Severity Scale score), C-GAS (Children’s Global Assessment Scale), WISC-IV (Wechsler Intelligence Scale for Children), and USCRS (UFMG Sydenham’s Chorea Rating Scale). ASD = autism spectrum disorder; PANS = pediatric acute-onset neuropsychiatric syndrome; R^2^ = coefficient of determination. Dots represent patients.

**Table 1 jcm-11-06493-t001:** Demographic data of the enrolled patients.

Classes	*n*	Female/Male	Age		
			Mean Value	SD	Range
PANS	34	10/24	9.1	2.90	5–16
ASD	15	0/15	9	4.28	3–17
Controls	25	9/16	12	2.17	8–17

PANS = pediatric acute-onset neuropsychiatric syndrome; ASD = autism spectrum disorder; SD = standard deviation.

**Table 2 jcm-11-06493-t002:** Statistical parameters of the multivariate models and the respective permutation test.

Models					
Models	R^2^X	R^2^Y	Q^2^	*p*-Value	Permutation Test: Intercept R^2^\Q^2^
ASD vs. Controls	0.49	0.61	0.49	0.0001	0.29/−0.22
ASD vs. PANS	0.41	0.51	0.25	0.02	0.32/−0.18

ASD = autism spectrum disorder; PANS = pediatric acute-onset neuropsychiatric syndrome; R^2^X, R^2^Y, Q^2^ = the variance and the predictive ability established to evaluate the strength of the models; *p*-value = probability value.

**Table 3 jcm-11-06493-t003:** Statistical parameters of the univariate analysis from the comparisons between PANS and ASD patients and ASD and controls. Mann–Whitney U-test and ROC curves were performed.

	Serum Samples							
	Metabolites	ASD	*p*-Value	*p*-Value Corrected	ROC-CURVE			
					AUC	Std. Error	CI	*p*-Value
ASD vs. Controls	Asparagine	−	0.006	0.06	0.77	0.07	0.6–0.9	0.007
	Aspartate	+	0.0008	0.1	0.82	0.07	0.7–0.9	0.001
	Betaine	+	0.02	0.06	0.71	0.08	0.5–0.9	0.02
	Glucose	−	0.03	0.01	0.66	0.08	0.5–0.8	0.09
	Glycine	−	<0.0001	0.1	0.96	0.02	0.9–1	<0.0001
	Lactate	+	0.02	0.03	0.72	0.08	0.5–0.8	0.02
	Pyruvate	−	0.04	0.1	0.70	0.08	0.5–0.9	0.04
ASD vs. PANS	Arginine	+	0.0003	0.004	0.82	0.06	0.7–0.94	0.0005
	Aspartate	+	0.002	0.02	0.82	0.06	0.7–0.95	0.001
	Betaine	+	0.001	0.005	0.71	0.08	0.54–0.88	0.01
	Choline	+	0.006	0.01	0.76	0.07	0.61–0.90	0.007
	Creatine Phosphate	−	0.01		0.74	0.07	0.59–0.88	0.01
	Glycine	−	<0.0001	0.005	0.85	0.05	0.74–0.96	0.0001
	Pyruvate	−	0.002		0.70	0.08	0.52–0.87	0.04
	Tryptophan	+	0.01	0.02	0.73	0.08	0.56–0.9	0.021

ASD = autism spectrum disorder; PANS = pediatric acute-onset neuropsychiatric syndrome; *p*-value = probability value; ROC-CURVE = receiver operating characteristic curve; AUC = area under the ROC-CURVE; Std. error = standard error; CI = confidence interval.

**Table 4 jcm-11-06493-t004:** Statistical results of the PLS correlation analysis.

Class	Clinical Parameters						
	PARS R^2^	PANSS R^2^	CYBOCS R^2^	YGTSS R^2^	C-GAS R^2^	TIQ R^2^	USCRS R^2^
PANS	0.4	0.7	0.4	0.2	0.3	0.2	0.3
ASD	0.5	0.6	0.5	-	0.8	0.8	0.9

Statistical parameters of the PLS correlation analysis between the complete metabolic profile of PANS and autism spectrum disorder patients and their corresponding psychodiagnostic scales scores. R^2^ statistic indicates the goodness-of-fit measure for linear regression models; that is, the percentage of variance in the dependent variable collectively explained by the independent variable. R^2^ measures the strength of the relationship between the independent variable (the complete metabolic profile in each group) and the dependent variable (psychodiagnostic scales scores) on a convenient 0–100% scale, where, in this case, 100% = 1. Footnote: PANS = pediatric acute-onset neuropsychiatric syndrome; PARS = Pediatric Anxiety Rating Scale; PANSS = Pediatric Acute Neuropsychiatric Symptom Scale; CYBOCS = Children’s Yale-Brown Obsessive Compulsive Scale; YGTSS = Total Yale Global Tic Severity Scale score; C-GAS = Children’s Global Assessment Scale; WISC-IV = Wechsler Intelligence Scale for Children; USCRS = UFMG Sydenham’s Chorea Rating Scale.

## Data Availability

The raw data supporting the conclusions of this article will be made available by the authors, without undue reservation.

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
