# Peer review of "^1^H-NMR-Based Metabolomics in Autism Spectrum Disorder and Pediatric Acute-Onset Neuropsychiatric Syndrome"

_jcm, 2022, doi:10.3390/jcm11216493_

Round 1

Reviewer 1 Report

I suggest the authors review the EQUATOR reporting guidelines (according to the study design) to organize the information correctly and not omit necessary details.

The ideas of the introduction must be organized so that there is a correct common thread and expand on the evidence supporting the possibility of markers in common between the two disorders. In addition, it should be clear to the reader the contribution to the clinical practice of this research.

The introduction's final part should contain the study's objective clearly and concisely. The display seems confusing.

The methodology is poorly structured and does not provide all the necessary information. For example, where was the sample recruited from, and has the sample size been calculated? The study design is also nowhere shown as well as any report on the data analysis. The "Participants" section contains information from the "Ethical Aspects" section. The variables measured are not explained in detail, nor is their measurement justified. It is recommended to follow a logical and clear structure for the reader: "Study design", "Participants", "Procedure", "Variables", "Statistical analysis", and "Ethical aspects".

Table 1 is challenging to understand if it is not explained in the methodology how categorical variables are described, usually by n and %, and quantitative variables, usually by median and interquartile range or mean and standard deviation, depending on the normality of the variable. This should be checked, and the test with which it is checked should be detailed. In addition, authors are advised not to include a mode in Table 1 but to include all the sociodemographic information of the participants since it can be influential in the analyses and should be collected and analyzed.

The results should be directly consistent with the study's objective(s), and in the current state of the manuscript, this cannot be verified.

The discussion is a section that should begin with a synthesis of the main results (not the objective) and then make a direct comparison with other similar studies and provide the reader with plausible explanations for the results. Unfortunately, the discussion of this work is not clear, mainly because the objectives are not clear.

The conclusion should not include comparisons with other studies. Instead, the conclusion should respond to the hypotheses of the research team.

The "Financing" section should be revised, and the recognized conflicts of interest are unclear to the reader.

The bibliography seems too extensive for the usual. It should be revised by selecting only recent articles.

Author Response

We would like to thank the reviewer which provided us with important suggestions to improve the manuscript.

1)     Q. I suggest the authors review the EQUATOR reporting guidelines (according to the study design) to organize the information correctly and not omit necessary details.

A. Thank you for this important suggestion. Checklists are not simply an administrative hurdle. Through the use of the STROBE Check-list for Observational and cohort studies, we tried to check that all of the important information has been included in our article

2)     Q. The ideas of the introduction must be organized so that there is a correct common thread and expand on the evidence supporting the possibility of markers in common between the two disorders. In addition, it should be clear to the reader the contribution to the clinical practice of this research.

A.     We modified the introduction section based on your suggestions.

3)      Q. The introduction's final part should contain the study's objective clearly and concisely. The display seems confusing.

A.     We described the study's objective more clearly as you suggested.

4)     Q. The methodology is poorly structured and does not provide all the necessary information. For example, where was the sample recruited from, and has the sample size been calculated? 41 The "Participants" section contains information from the "Ethical Aspects" section. The variables measured are not explained in detail, nor is their measurement justified. It is recommended to follow a logical and clear structure for the reader: "Study design", "Participants", "Procedure", "Variables", "Statistical analysis", and "Ethical aspects".

A.       As described in the first paragraph of the method section, patients and controls were enrolled at the outpatient service of the Child and Adolescent Neuropsychiatric Unit, “G. Brotzu” Hospital Trust, Cagliari. We couldn’t perform the calculation of the sample size because usually, this type of analysis requires a number of different parameters to determine the minimum number of subjects that need to be enrolled in a study in order to have sufficient statistical power. For example, the baseline incidence or the population variance. Especially for PANS disease (which is a relatively “new” neuropsychiatric syndrome that often overlaps with other diseases such as Tourette syndrome, Obsessive-Compulsive disorder etc), epidemiological data are still under continuous update and actually is not possible to refer to incidence and variance between the population.  Furthermore, many researchers in different countries and regions do not recognize PANS as a disease.

We modified the structure of the methods section as follows: “Study design" (where we included the kind of the analysed variables), "Participants and Ethical aspects", “Psychiatric evaluation”. The paragraph “Sample Preparation and data analysis” contains the link or the reference to the supplementary material where you can find all the details about "Procedure", "Statistical analysis", etc.

5) Q. Table 1 is challenging to understand if it is not explained in the methodology how categorical variables are described, usually by n and %, and quantitative variables, usually by median and interquartile range or mean and standard deviation, depending on the normality of the variable. This should be checked, and the test with which it is checked should be detailed. In addition, authors are advised not to include a mode in Table 1 but to include all the sociodemographic information of the participants since it can be influential in the analyses and should be collected and analyzed.

A. We added a brief description of the variables in the text and deleted the mode and the median from Table 1. We avoided adding further sociodemographic information (e.g. scholarity, urban or extra-urban  living, family numerousity, family economic level of) in order not to add further variables which could increase the complexity of the paper and their correct interpretation. We applied the D’Agostino & Person normality test which evidenced a normal distribution of the values. 

6) Q. The results should be directly consistent with the study's objective(s), and in the current state of the manuscript, this cannot be verified.

A. We modified the result section based on your suggestions.

7) The discussion is a section that should begin with a synthesis of the main results (not the objective) and then make a direct comparison with other similar studies and provide the reader with plausible explanations for the results. Unfortunately, the discussion of this work is not clear, mainly because the objectives are not clear.

A. We modified the discussion section trying to make it more exhaustive

8) The conclusion should not include comparisons with other studies. Instead, the conclusion should respond to the hypotheses of the research team.

A. We modified the conclusion section adding some limitations of the study.

9) The "Financing" section should be revised, and the recognized conflicts of interest are unclear to the reader.

A. We corrected the section of the Funding.

The bibliography seems too extensive for the usual. It should be revised by selecting only recent articles.

A. We are aware of the high number of references, but they are all required to corroborate the results and comments. 

Reviewer 2 Report

This research is of great interest to researchers of childhood-onset autism, or pediatric ASD. This research is of great interest to researchers of childhood-onset autism, or ASD, because while there are many theories about the biomarkers characteristic of autism, none of them have yet reached a dominant conclusion. Researchers around the world are currently using various methods to study markers in children with autism, and this paper is one of those challenges.
This paper analyzes the profiles of various substances in normal controls using plasma metabolomics analysis techniques.
Using the same approach, we also analyze the plasma metabolomics profile of PANS, a neuropsychiatric syndrome with acute onset in childhood.
The results describe characteristic trends in substances such as asparagine, aspartate, betaine, glycine, lactate, glucose, and pyruvate for the normal and ASD groups. In addition to the substances mentioned above, the PANS and ASD groups were characterized by tryptophan and other substances. In the PANS and ASD groups, the characteristics of tryptophan, etc., in addition to the substances mentioned above, are described.
The mechanisms of each are discussed, including effects on glutaminergic and dopaminergic downstream pathways, as well as on GABA-mediated transmission of molecular substances. In addition, PANS is discussed in relation to NMDA receptor dysfunction. PANS is also discussed in relation to NMDA receptor dysfunction.

In order to make the content clearer to the reader, I would like to request the following major additions.

Major 1
The patients with PANS in this paper were diagnosed by a child psychiatrist. However, while ASD is an internationally agreed upon concept, many researchers in different countries and regions do not recognize PANS as a disease concept.
The article describes PANS as a psychiatric disorder such as childhood oppositional defiant disorder with acute onset caused by infectious or immune-mediated agents, but I would like to know more specifically what the authors recognize as PANS, what viruses, bacteria, fungi, etc. are most often associated with PANS, and what are the consequences of such an association. What are the immunological abnormalities that result in the clinical manifestations associated with NMDAR?

Major 2
How can we understand the profile obtained in clinical practice? Is it a biomarker for clinical diagnosis or is it useful for disease typing? Should they be understood in relation to understanding the severity of the disease?

Major 3
At least in current medicine, ASD is considered an inherited condition. In addition, PANS is unknown. Do these profile abnormalities reflect congenital abnormalities? In other words, are they microscopic inborn errors of metabolism? Or are they acquired consequences? We would appreciate any additional comments from the authors.

Major 4
If possible, please discuss whether continued analysis of these profiles will open the possibility of new drug discovery in pediatric autism. Please add the authors' opinions to the discussion section. Or, if there are any papers on such drug discovery in autism, please add them.

Overall, the argument of this paper is clear and worthy of JCM. We have requested four further detailed comments and would like to know the authors' thoughts on the subject for better reader comprehension.

Best regards, Dr. Reviewer

Author Response

We would like to thank the reviewer which provided us with important suggestions to improve the manuscript.

This research is of great interest to researchers of childhood-onset autism, or pediatric ASD. This research is of great interest to researchers of childhood-onset autism, or ASD, because while there are many theories about the biomarkers characteristic of autism, none of them have yet reached a dominant conclusion. Researchers around the world are currently using various methods to study markers in children with autism, and this paper is one of those challenges.
This paper analyzes the profiles of various substances in normal controls using plasma metabolomics analysis techniques.
Using the same approach, we also analyze the plasma metabolomics profile of PANS, a neuropsychiatric syndrome with acute onset in childhood.
The results describe characteristic trends in substances such as asparagine, aspartate, betaine, glycine, lactate, glucose, and pyruvate for the normal and ASD groups. In addition to the substances mentioned above, the PANS and ASD groups were characterized by tryptophan and other substances. In the PANS and ASD groups, the characteristics of tryptophan, etc., in addition to the substances mentioned above, are described.
The mechanisms of each are discussed, including effects on glutaminergic and dopaminergic downstream pathways, as well as on GABA-mediated transmission of molecular substances. In addition, PANS is discussed in relation to NMDA receptor dysfunction. PANS is also discussed in relation to NMDA receptor dysfunction.

In order to make the content clearer to the reader, I would like to request the following major additions.

Major  1
The patients with PANS in this paper were diagnosed by a child psychiatrist. However, while ASD is an internationally agreed upon concept, many researchers in different countries and regions do not recognize PANS as a disease concept.
The article describes PANS as a psychiatric disorder such as childhood oppositional defiant disorder with acute onset caused by infectious or immune-mediated agents, but I would like to know more specifically what the authors recognize as PANS, what viruses, bacteria, fungi, etc. are most often associated with PANS, and what are the consequences of such an association. What are the immunological abnormalities that result in the clinical manifestations associated with NMDAR?

A: Gromark and coll. (2019) described the frequency of onset symptoms of a PANS cohort and confirmed obsession and compulsion (89%) to be the main symptoms at outset, followed by anxiety (including separation anxiety) (78%), emotional lability and/or depression (71%), sleep disorders (69%), and complex tics (62%). Very common (60%) were other motor abnormalities such as choreic movements, dystonia, muscle weakness, difficulties with gross and fine motor skills. Cognitive symptoms, such as attention deficit (63%), hyperactivity (43%) deterioration in school performance (50%) were also frequently Less commonly seen, at least at onset, were irritability/aggression (44%) and regressive behaviors (40%). Eating disorder (significant loss of appetite resulting in weight loss, avoidant–restrictive food intake disorder, or other OCD-related eating disorder) affected 40% of subjects. In addition to symptoms listed above, from 14% up to 37% of PANS patients can present with psychotic symptoms such as auditory and/or visual hallucinations, thought disorganization, and delusions. Patients who experience psychotic symptoms tend to have a more severe long-term impairment (Silverman et al, 2019).

As for the associated pathogens, PANS symptoms can occur after the exposure to a wide variety of infections other than Streptococcus, e.g. bacterial (M. pneumoniae, B. burgdorferi, S. aureus) and viral (Epstein–Barr, Influenza, Coxsackie, Varicella, SARS-CoV2), as well as to non-infectious environmental triggers such as oxidative or emotional stress, and toxin exposure (Sweedo et al, 1997; Calaprice et al, 2017).

Interestingly, psychosis is a common feature of the pediatric autoimmune encephalitidies (AE), such as anti-NMDAR and Hashimoto's encephalopathathy. Interestingly, AE may occur with a wide range of symptoms shared with PANS, such as abnormal movements, cognitive and memory impairment, behavioral changes, and autonomic dysregulation. Other differences between PANS and AE, such as the rarity of seizures and cognitive impairment in PANS or the spontaneous relapsing-remitting course in PANS, differently from AE, where symptoms progress rapidly to a maximum severity level and resolution is uncommon without treatment (Cellucci et al, 2020). Nevertheless, these differences seem to be more quantitative than qualitative and, in principle, PANS and AE could be considered two phenotypes belonging to the same autoimmune spectrum.

Major 2
How can we understand the profile obtained in clinical practice? Is it a biomarker for clinical diagnosis or is it useful for disease typing? Should they be understood in relation to understanding the severity of the disease?

A. Thank you for the question. We applied the metabolomics approach that, in general, allows us to identify, when possible, specific patterns of biomolecules (metabolites) which could be specific for a given pathological state. The identified metabolites could be useful for mainly two purposes: to improve the diagnostic path of a disease where the diagnosis is still challenging, or to improve knowledge in terms of biological/pathophysiological mechanisms still unclear in a given disease. The latter point could be beneficial for example to find or discover new therapeutic targets etc.

On the other hand, metabolomics is still a young science, and for this reason, related technologies and statistical tools are constantly evolving making difficult the quick application directly to clinical practice. With our study, we have been tracing the way for a further and deeper investigation as we obtained promising preliminary results.

Major 3
At least in current medicine, ASD is considered an inherited condition. In addition, PANS is unknown. Do these profile abnormalities reflect congenital abnormalities? In other words, are they microscopic inborn errors of metabolism? Or are they acquired consequences? We would appreciate any additional comments from the authors.

A: Thanks for this comment. It is important to assume a reasonable pathogenic model for PANS. Currently, it could be based on the assumption that genetic risk factors, leading to dysregulation of immune pathways, could play a role in PANS. In particular, some genetic subgroups could have specific markers of inflammation or autoantibodies leading to a vulnerability to develop the disorder. The whole exome sequencing (WES) and the whole genome sequencing (WGS) have been very recently used to identify biologically powerful genetic factors underlying PANS (Trifiletti et al, 2022). The study identified ultra-rare variants in 11 genes (PPM1D, SGCE, PLCG2, NLRC4, CACNA1B, SHANK3, CHK2, GRIN2A, RAG1, GABRG2, and SYNGAP1) in 21 PANS subjects. The study is the first demonstration that de novo or ultra-rare deleterious genetic variants in PANS regulate multiple levels of the neuroinflammatory circuit (from peripheral and central innate immunity, to synaptogenesis) and could interfere with the function of the blood-CSF barrier, as well as with the enteric nervous system. The authors described two broad functional categories of genes regulating the peripheral immune responses and microglia (PPM1D, CHK2, NLRC4, RAG1, PLCG2) and the synaptogenesis (SHANK3, SYNGAP1, GRIN2A, GABRG2, CACNA1B, SGCE). These last genes are involved in the pathogenesis of neurodevelopmental disorders and, in particular, of autism spectrum disorder (ASD) (Leblond et al, 2014), suggesting a multifactorial and probabilistic role of genetic ad environmental factors (different kinds of stressors such as infections) both in PANS and in ASD. We have added a brief comment on this issue in the introduction. 

Major 4
If possible, please discuss whether continued analysis of these profiles will open the possibility of new drug discovery in pediatric autism. Please add the authors' opinions to the discussion section. Or, if there are any papers on such drug discovery in autism, please add them.

A: Hopefully, these kind of studies will lead, in the next future, to discovery new treatment options. This is a crucial point and we thank the reviewer for having stress it.  Our study is consistent with previous studies on amino acid dysregulation designed to stratifying ASD based on “metabotypes” (Smith et al, 2019).  Both for ASD and PANS, the identification of “metabotypes” is the first step of a process that could lead to arrange metabolic tests and new targeted therapeutic interventions. We have stated this  in the conclusion section.

Overall, the argument of this paper is clear and worthy of JCM. We have requested four further detailed comments and would like to know the authors' thoughts on the subject for better reader comprehension.
Best regards, Dr. Reviewer

A: We would like to thank the reviewer for his very useful questions and comments. We strongly think that, following his suggestions, we have improved the global quality of our paper. Furthermore, the proposed subjects have improved the usability of the information for the readers.

Round 2

Reviewer 2 Report

The authors submitted comments on the scientific answers to four major questions from the reviewers. They also revised the paper to better reflect the comments. The paper will be a very influential paper for readers.

The resolution of the figures and graphs is not clear enough for publication, possibly due to the number of pixels on PC screens. Fine lines etc. cannot be seen well. Reviewers will be asked to resubmit clearer images.

Best regards,

Dr. Reviewer

Author Response

Dear Reviewer,

thank you for your suggestion.

I have "regenerate" the figures in TIFF with 300dpi.